# Implementation Strategies Used to Increase Human Papillomavirus Vaccination Uptake by Adolescent Girls in Sub-Saharan Africa: A Scoping Review

**DOI:** 10.3390/vaccines11071246

**Published:** 2023-07-16

**Authors:** Mwansa Ketty Lubeya, Mulindi Mwanahamuntu, Carla J. Chibwesha, Moses Mukosha, Mercy Wamunyima Monde, Mary Kawonga

**Affiliations:** 1Department of Obstetrics and Gynaecology, School of Medicine, The University of Zambia, Lusaka 10101, Zambia; mulindim@gmail.com; 2Women and Newborn Hospital, University Teaching Hospitals, Lusaka 10101, Zambia; 3School of Public Health, Faculty of Health Sciences, University of the Witwatersrand, Johannesburg 2193, South Africa; mukoshamoses@yahoo.com (M.M.); mary.kawonga@wits.ac.za (M.K.); 4Clinical HIV Research Unit, Helen Joseph Hospital, Johannesburg 2193, South Africa; carla_chibwesha@med.unc.edu; 5Department of Pharmacy, School of Health Sciences, The University of Zambia, Lusaka 10101, Zambia; 6School of Medicine Library, The University of Zambia, Lusaka 10101, Zambia; mercy.wamunyima@unza.zm; 7Department of Community Health, Charlotte Maxeke Johannesburg Academic Hospital, Johannesburg 2193, South Africa

**Keywords:** HPV vaccine uptake, cervical cancer elimination, HPV immunization, implementation strategies, feasibility, scoping review, ERIC taxonomy, adolescent HPV vaccination, Africa

## Abstract

Barriers to successful implementation of the human papillomavirus vaccination exist. However, there is limited evidence on implementation strategies in sub-Saharan Africa (SSA). Therefore, this scoping review aimed to identify implementation strategies used in SSA to increase HPV vaccination uptake for adolescent girls. This scoping review was guided by Joanna Briggs Institute guidelines for scoping reviews and an a *priori* protocol and reported based on the Preferred Reporting Items for Systematic Reviews and Metanalysis for Scoping Reviews (PRISMA-ScR). We searched PubMed, EMBASE, CINAHL, Scopus, Google Scholar, and gray literature. Two independent reviewers screened article titles and abstracts for possible inclusion, reviewed the full text, and extracted data from eligible articles using a structured data charting table. We identified strategies as specified in the Expert Recommendation for Implementing Change (ERIC) and reported their importance and feasibility. We retrieved 246 articles, included 28 of these, and identified 63 of the 73 ERIC implementation strategies with 667 individual uses, most of which were highly important and feasible. The most frequently used discrete strategies included the following: Build a coalition and change service sites 86% (24/28), distribute educational materials and conduct educational meetings 82% (23/28), develop educational materials, use mass media, involve patients/relatives and families, promote network weaving and stage implementation scale up 79% (22/28), as well as access new funding, promote adaptability, and tailor strategies 75% (21/28). This scoping review shows that implementation strategies of high feasibility and importance were frequently used, suggesting that some strategies may be cross-cutting, but should be contextualized when planned for use in any region.

## 1. Introduction

The human papillomavirus (HPV) primarily affects human beings and is one of the most common sexually transmitted agents affecting approximately 3 out of 4 sexually active people early in their sexual life [1,2]. Generally, the point prevalence of HPV infection is estimated at 11–12% globally and 22–24% for sub-Saharan Africa (SSA) [3,4,5,6]. There are several strains of HPV (>200) categorized into low-risk and high-risk types [7].

Cancers caused by HPV include anal, penile, oropharyngeal, vulval, and the commonest being cervical cancer, which has the highest incidence in SSA [4]. Paradoxically, SSA has low cervical cancer screening uptake [8,9] and delayed diagnosis and treatment [10]. Furthermore, women living with the Human Immunodeficiency Virus (HIV) are more likely to have persistent HPV infections, with more rapid progression to malignancies [11], even in women on antiretroviral therapy [12,13,14].

Primary prevention of cervical cancer is possible with the HPV vaccination of prepubertal girls aged 9 to 14 years before sexual debut [2]. Currently (2023), the available licensed HPV vaccines are classified as bivalent, quadrivalent, and non-valent based on the number of targeted HPV strains [15]. Evolving evidence has shown that fewer doses are effective, prompting the World Health Organization (WHO) to recommend one to two doses for girls aged 9–20 years and one to two doses for those aged 21 years or older with a 6-month interval [16,17]. The three-dose schedule is indicated in immunocompromised adolescents, such as those living with HIV, due to limited evidence in this population [15,17]. The reduced number of doses shows promise in increasing HPV vaccine uptake by mitigating barriers, such as multiple visits to the health facility, loss to follow-up, cost of the vaccine and its delivery, and human resource [18,19,20,21]. 

The benefits of HPV vaccination are evident in high-income countries which introduced national HPV vaccination programs earlier. A meta-analysis including fourteen high-income countries implementing the HPV vaccination program reported a reduction in the prevalence of HPV 16 and 18 among adolescent girls and young women [22]. Similarly, the United Kingdom shows that cervical cancer has almost been eliminated after 10 years of the national HPV vaccination of adolescent girls [23]. 

On the contrary, many SSA countries are yet to implement national HPV vaccination programs and further challenged by low vaccine uptake in implementing countries [18,24]. Some identified barriers to HPV vaccine uptake include individual, structural, economic, community/social, and cultural issues [20,25] and, recently, the COVID-19 pandemic [26]. Furthermore, systemic health system constraints occur in many SSA countries, including low levels of knowledge among various stakeholders, financing, vaccine communication, and community engagement [27,28]. 

There is an urgent need at the global level to increase HPV vaccine uptake, a priority of the WHO accelerated global strategy for cervical cancer elimination, with targets set for 2030 [29]. The WHO strategy has the following triple targets; 90% of girls by the age of 15 years should be fully vaccinated with the HPV vaccine, 70% of women by age 35 years should be screened with a high-performance screening test and again screened at 45 years of age and 90% of women with pre-cancer treated and 90% of invasive carcinoma managed properly [29].

Considering the long-standing need to increase HPV vaccination coverage in SSA and now with the pronouncement of WHO cervical cancer elimination strategy to achieve success in different real-world settings, implementation strategies are necessary to overcome bottlenecks and reach the target groups. Implementation strategies are “a systematic intervention process to adopt and integrate evidence-based health innovations into usual care” [30]. Some examples of implementation strategies are to; distribute educational materials, build a coalition, conduct educational meetings, change service sites, and obtain and use consumer feedback [31,32,33]. The need to better understand and evaluate the implementation of evidence-informed HPV vaccination programs in high-burden regions, such as SSA has been recognized in policy statements [29] and synthesized literature [34]. Therefore, it is important to know the breadth and depth of ongoing implementation strategies in SSA used to increase HPV vaccination uptake for cervical cancer prevention. 

A systematic review by Mavundza et al. [34] on effective interventions to increase HPV vaccine uptake was limited to controlled randomized trials performed in high-income countries despite having the primary goal of reviewing data from all income brackets. Studies from low- and middle-income countries did not meet the inclusion criteria as most were considered poorly designed. We conducted a preliminary systematic search of online databases, which showed that no recent systematic or scoping review has been undertaken or was underway in SSA to describe implementation strategies used to increase HPV vaccine uptake and their effectiveness. We, therefore, elected to conduct a scoping review as an ideal methodological approach to achieve our aim [35] to identify and map the identified implementation strategies used in SSA for increasing HPV vaccination for adolescent girls according to the compilation of Expert Recommendation for Implementation Change (ERIC) [36] and the corresponding nine clusters [37] as a starting point. 

**Research question:** What implementation strategies have been used to increase human papillomavirus vaccination by adolescent girls in sub-Saharan Africa? 

## 2. Materials and Method

This scoping review followed the Joanna Briggs Institute (JBI) updated methodology for scoping reviews [38] and an *a priori* protocol [39]. 

### 2.1. Inclusion Criteria

#### 2.1.1. Participants

We included studies aiming to increase HPV vaccine uptake by adolescent girls; however, these implementation strategies often target various stakeholders. Therefore, participants included adolescent girls, parents/guardians, healthcare providers, teachers, key community leaders, politicians, and policymakers. 

#### 2.1.2. Concept

We included studies that used at least one of the 73 implementation strategies defined by the ERIC taxonomy [36] to increase the uptake of HPV vaccination by adolescent girls. 

The second step of the ERIC compilation validated the distinctness of the 73 strategies and organized them into nine clusters: 1. Use evaluative and iterative strategies, 2. provide interactive assistance, 3. adapt and tailor to context, 4. develop stakeholder interrelationship, 5. train and educate stakeholders, 6. support clinicians, 7. engage consumers, 8. utilize financial strategies, and 9. change infrastructure [37]. The 73 strategies were later rated based on relative importance and feasibility and supported by the formation of go-zone quadrants based on perceived feasibility and importance: (I) Strategies with high importance and feasibility, (II) high feasibility low importance, (III) low importance and feasibility, and (IV) high importance low feasibility according to Waltz et al. [37]. 

#### 2.1.3. Context

This review considered only studies carried out in SSA aimed at increasing the uptake of the HPV vaccine by adolescent girls. 

### 2.2. Types of Sources

We considered all study designs, peer-reviewed studies, gray literature, and reports. Narrative literature reviews and articles including multiple countries were included if they had country-specific data for any SSA country and met the inclusion criteria [39]. We read through all the articles mentioning interventions to check which ones met the definitions by the ERIC taxonomy. 

### 2.3. Search Strategy

We initially searched PubMed, EMBASE, and Scopus to identify articles on adolescent HPV vaccination, guided by a medical librarian. The text words in the titles and abstracts of relevant articles and the index terms used to describe the articles were used to develop a full search strategy for PubMed (Appendix A). This search strategy, with all identified keywords and index terms, was adapted for each included database and information source: PubMed, EMBASE, Cumulative Index to Nursing and Allied Health Literature (CINAHL), Google Scholar, and Scopus. Gray literature citations (unpublished, non-peer review data sources including conference proceedings, abstracts, and meetings) were reviewed with other peer-reviewed publications using the same inclusion and exclusion criteria. 

The reference lists of included sources of evidence were screened for additional studies. Similar articles in PubMed were also searched. Studies published in other languages besides English were excluded due to the teams’ limited competence. We included studies published from January 2006, when WHO approved the use of the HPV vaccine among prepubertal girls through December 2021. 

### 2.4. Sources of Evidence Selection

Data from identified articles were collated and uploaded into Endnote X9 (Clarivate Analytics, Philadelphia, PA, USA), removed duplicates, and imported the rest of the studies into the JBI System for the Unified Management, Assessment and Review of Information (JBI SUMARI: JBI, Adelaide, Australia) [40]. Two independent reviewers (MKL and MM) screened titles and abstracts and assessed the full text of selected citations in detail against the inclusion criteria. Full-text studies and articles that did not meet the inclusion criteria were excluded. Reasons for exclusion are provided in Figure 1. Any disagreements between the reviewers at each stage of the study selection process were resolved through discussion or with a third reviewer (any of MK, CC, or MM#) if consensus was not reached.

Sources of information for the implementation strategies included surveys within vaccination programs or Expanded Programs for Immunization managers, government documents, community surveys, registers for HPV vaccinations, HPV training manuals, micro plans, training reports, and support supervision reports. The search results are presented in a Preferred Reporting Items for Systematic Reviews and Meta-Analyses Extension for Scoping Review (PRISMA-ScR) flow diagram [41] in Figure 1. 

### 2.5. Data Extraction and Charting

Data from the included papers were extracted and summarized by two independent reviewers (MKL and MMW) and checked by MK using a data extraction chart developed by this team. The extracted data included the first author’s name, year of publication, country of study, the title of the article, article type, study design (where applicable), targeted stakeholders, and type of program (national program/demonstration/subpopulation).

The draft extraction chart was trialed on five included papers by (MKL, MMW, and MM). The revisions included (i) clarifying the sample size to be that of the eligible girls other than study participants as this was dynamic and not consistently reported; (ii) not to report the framework for data collection as this was rarely reported.

A meeting was held among MKL, MK, MWM, and MM to pilot the coding process of the ERIC strategies on three included studies. Three reviewers (MKL, MWM, and MM) independently coded and charted data for the specific ERIC implementation strategies and their allocated codes (1 to 73) along with the corresponding ERIC cluster (1 to 9). Co-authors MKL and MWM iteratively and inductively coded other strategies which emerged but could not neatly fit into the ERIC compilation [42]; these were verified by MK. 

### 2.6. Data Analysis and Presentation

The extracted data are presented in table form to align with the study objectives and accompanied by a narrative summary. The identified strategies were mapped onto the ERIC compilation, grouped into the ERIC’s nine conceptually relevant clusters, and mapped to the four go-zones to understand their relative importance and feasibility rating according to Waltz et al. [37]. This scoping review is written according to the PRISMA-ScR checklist (Appendix A).

## 3. Results

### 3.1. Characteristics of Included Studies

Our search yielded 246 articles and finally included 28 studies as shown in the Prisma flow chart in Figure 1. Most articles included multiple countries *n* = 8 [43,44,45,46,47,48,49,50], while the rest included single countries as follows: South Africa *n* = 3 [51,52], Malawi *n* = 2 [53,54], Tanzania *n* = 2 [55,56], Uganda *n* = 2 [57,58], and one each for Botswana [59], Cameroon [60], The Gambia [61], Ghana [62], Mozambique [63], Nigeria [64], Rwanda [65], Senegal [66], Togo [67], Zambia [68], and Zimbabwe [69]. 

A significant amount of data was from demonstration projects, *n* = 14 [43,44,45,46,50,51,54,57,58,61,63,67,68,70] followed by other sub-national research projects *n* = 7 [31,52,53,55,60,62,64], national programs *n* = 5 [48,56,65,66,69], and two had mixed data sources [47,49]. Of the national vaccination programs identified, 26 had international financial support to acquire the vaccines delivered free of charge to eligible girls. Half of the reported funding was from Gavi, the vaccine alliance (14/28). Most of the included articles were research papers *n* = 18 [31,43,44,45,47,48,49,51,52,53,54,55,57,60,62,66,69], followed by reports *n* = 5 [56,58,59,61,68], evaluations *n* = 3 [46,63,67], and perspectives *n* = 2 [50,65]. Publication year ranged from 2011 to 2021 with no articles identified for 2022, 2019, and 2014. About a third of the studies *n* = 8 were published in 2021 [48,49,53,56,62,66,67,69], *n* = 2 in 2018 [31,63], *n* = 3 in 2017 [45,54,64], *n* = 2 in 2016 [47,50], *n* = 4 in 2015 [52,57,61,70], *n* = 3 in 2013 [44,51,68], *n* = 4 in 2012 [46,55,60,65], and *n* = 2 in 2011 [43,58]. See Table 1 for details.

### 3.2. Characteristics of the Eligible Adolescent Girls

The target adolescents’ age group varied from 9 to 18 years or grades four to seven. One study used the recommended multi-age cohort outside demonstration projects [69]. Other studies for countries introducing national programs used a single-age cohort of 9-year-old girls [66] or 14-year-old girls [56].

### 3.3. Vaccine Delivery Approaches

Some studies indicated reaching out-of-school girls through community sensitization and active tracing by community healthcare workers [48,54,56,63,65,68]. Other studies only focused on school delivery approaches [31,59]. In some studies, there were multiple locations where girls could access the vaccine, such as from schools and health facilities for out-of-school girls or school girls who missed the vaccine offered at school [48,50,56,65]. In some instances, HPV vaccination delivery was integrated with other health programs, such as deworming, vitamin A supplementation, nutrition counselling, infectious diseases prevention, tetanus toxoid vaccination, and other sexual health reproductive education [43,54,56,65,67].

### 3.4. Implementation Strategies According to ERIC Compilation

We identified 63 of the 73 ERIC implementation strategies across the 28 studies, while 10 were not reported in any article (Appendix A). All included studies reported using multifaceted implementation strategies, which indicates combining two or more strategies. The most commonly reported discrete strategies were to build a coalition *n* = 24 [43,45,46,48,49,50,51,52,54,56,58,59,61,63,65,66,67,68,69,70], change service sites *n* = 24 [31,43,45,46,47,48,49,50,51,52,54,56,57,58,59,60,61,63,65,66,67,68,69,70], conduct educational meetings *n* = 23 [31,43,44,45,46,47,48,49,50,51,52,56,58,59,62,63,64,65,66,67,68,69,70], distribute educational materials *n* = 23 [31,43,44,45,46,47,48,50,51,52,56,57,58,59,60,62,63,64,65,66,67,69,70], develop educational materials *n* = 23 [31,43,44,45,46,47,48,50,51,52,53,56,57,58,59,60,63,64,65,66,67,69,70], promote network weaving *n* = 22 [31,43,44,45,46,47,48,49,50,51,54,56,57,58,59,61,63,65,66,67,69,70], stage implementation scale up *n* = 22 [31,43,44,45,46,47,48,51,52,54,56,57,58,59,61,63,65,66,67,68,69,70], use mass media *n* = 22 [31,44,45,46,47,48,49,51,53,56,57,58,59,63,64,65,67,68,69,70], tailor strategies *n* = 21 [31,44,45,46,47,48,49,50,51,53,54,56,57,58,59,63,64,66,67,68,69], and access new funding (*n* = 21) [31,43,45,46,48,49,50,51,52,54,56,58,59,61,63,65,66,67,68,69,70]. Other strategies were coded inductively as they were used frequently but could not neatly fit into the ERIC taxonomy, such as investment in a system to report and manage Adverse Events Following Immunization (AEFI) and applying an opt-out approach to parental consent to vaccinate daughters. 

Table 2 shows the distribution of the identified implementation strategies within the nine ERIC clusters and the frequency of their use within the articles [37], this is detailed in Appendix A. All strategies within cluster 1 (use evaluative iterative and strategies), cluster 2 (provide interactive assistance), cluster 5 (Train and educate stakeholders), and cluster 7 (Engage consumers) were used at least once. Majority of the reported strategies of high importance and feasibility categories were within cluster 1: use evaluative and iterative strategies; cluster 4: develop stakeholder interrelationships; and cluster 5: train and educate stakeholders.

### 3.5. Targeted Stakeholders for Implementation Strategies

Stakeholders involved were varied within the included studies. Most articles included more than one stakeholder group, e.g., policymakers, politicians, healthcare workers, community volunteers, teachers, parents, and adolescent girls *n* = 23 [31,43,45,46,47,48,49,50,51,52,53,54,56,57,58,59,61,63,65,66,67,68,69], while others focused on exclusively specific stakeholders, such as parents *n* = 3 [45,60,62], healthcare workers *n* = 2 [63,64], or adolescents *n* = 1 [55]. 

## 4. Discussion

This scoping review aimed to identify implementation strategies to increase HPV vaccination of adolescent girls in SSA and map them according to the ERIC compilation. Majority of the identified implementation strategies were of high importance and feasibility falling within ERIC clusters 1, 4, and 5. Furthermore, most studies reported multifaceted implementation strategies which combine discrete strategies, and none neatly fit into blended strategies, which have been described as “multiple strategies packaged as a protocolized or branded implementation intervention” [30].

Strategies under the cluster of train and educate stakeholders were among the most frequently used. However, there is mixed evidence on the value of education strategies to increase the uptake of interventions generally and for HPV vaccination [71]. A recent systematic review carried out in India to measure the impact of educational interventions identified three studies focusing on educational interventions, and only one of these showed an increase in HPV vaccine uptake [72]. Contrary to this, a systematic review conducted in the USA reported that education by authoritative sources to parents increased HPV vaccine uptake by youths [73]. This is not surprising as knowledge has also been implicated as one of the barriers to HPV vaccination, suggesting that educational interventions may be of value if combined with other strategies and contextualized to a specific population. 

We recently demonstrated that knowledge levels in a population of parents in Zambia affected their consent to vaccinate their adolescent daughters against HPV [28]. Additionally, an interventional study using educational presentations to teachers, healthcare workers, and parents on various aspects of HPV and HPV vaccines found that increased knowledge increased vaccine uptake and acceptance of the HPV vaccine [74]. Therefore, a systematic review on educational intervention and knowledge levels among different stakeholders will be key to understanding the value of education-related interventions in the HPV vaccination space in SSA to change the current landscape. 

We found that multiple stakeholders worked together across the different studies, which was under the strategy “build a coalition,” cluster to develop stakeholder’s interactions. For example, healthcare workers played a pivotal role in administering the vaccine, and thus ought to be knowledgeable and have positive attitudes toward the vaccine [75], as their recommendation plays a crucial role in vaccine uptake across many geographical areas [76,77]. Teachers, as well, served as a conduit between the implementers and the user side in terms of information sharing and obtaining consent from the parents and assent from the girls. Ministries of health and education, including vaccination bodies, non-governmental organisations and the communities worked together to achieve a common goal of increasing HPV vaccination. Understanding the stakeholder landscape and networking during the planning and implementation of HPV vaccination programs is key to successful implementation. 

As expected, given the scope of our review, we did not find the strategies that SSA countries used to navigate the HPV vaccination roll-out during the COVID-19 pandemic. To date, the available evidence shows that some LMICs rolled out national HPV vaccination programs during the COVID-19 pandemic. In contrast some SSA countries, such as Cameroon, had their HPV vaccine introduction programs interrupted during the same period [48]. Therefore, data are urgently needed to halt the drop in vaccine uptake experienced, negating the gains made earlier. However, it is unclear if the implementation strategies could be applied to the SSA context, where the HPV uptake dropped due to school closures and misinformation around the COVID-19 pandemic [26]. 

Generally, there were fewer articles reporting implementation strategies for national HPV vaccination rollout programs despite reports of 17 countries currently implementing these programs [8]. There is a need for these data close to real-life settings considering that over 55 demonstration projects have already been carried out [78]. 

The change in service site which falls within the “change infrastructure” cluster was one of the most frequently used strategies. This was mostly seen as delivering HPV vaccines in schools outside the usual health facility settings. Most studies acknowledged that large numbers of adolescents targeted for the vaccine are found in schools, favouring this platform [43,48,68]. Additionally, combining strategies in the school vaccination program has been found to be useful. For example, a study conducted in the United States of America on interventions to increase HPV vaccination reported that school-based vaccination combined with community education of parents and adolescents showed a significant increase in the intervention group [79]. Other studies have even intimated that schools should be used as primary vaccination sites due to vast experience and proof of success [48,80]. 

However, school-based delivery poses a challenge for reaching out-of-school girls who may be more vulnerable to early sexual debut and exposed to HPV infection in early discordant marriages [81]. Some studies we identified in this scoping review did not include out-of-school girls in their demonstration programs [59]. Finding out-of-school girls in the face of heavily school-based HPV vaccination continues to be challenging, and more literature and experiences should be shared to inform policy. 

The most frequently used strategies under the cluster “adapt and tailor strategies” were to promote adaptability and tailor strategies. A systematic review and meta-analysis on effective communication strategies to increase HPV vaccination uptake among adolescent girls in SSA reported that tailoring communication interventions to specific stakeholders yielded positive results [82]. The review reported that the most effective communication strategies included those which educate the population about the HPV vaccine, facilitate decision-making on vaccine uptake and community ownership of the vaccination process.

Therefore, promoting, and tailoring strategies in the use of the HPV vaccine in SSA implies developing tactics specific to the region’s circumstances and problems. Addressing issues, such as culturally appropriate communication, scarce resources, inadequate healthcare facilities, and low knowledge is part of this. To guarantee that HPV vaccination is successful and ongoing, adaptation needs to be encouraged.

We identified monitoring of side effects as an implementation strategy outside the ERIC compilation. HPV vaccination can result in transient minor side effects, such as pain and redness on the injection site. This scoping review found that active management of side effects was used in some settings to increase HPV vaccination. Occurrences, such as mass hysteria among vaccine-eligible adolescents have been erroneously associated with the HPV vaccination [82]. These unfounded reported side effects have resulted in some countries, such as Japan and Cambodia to have very low HPV vaccine uptake compared to the initial levels when the programs were launched [83,84]. Effectively dealing with adverse events is crucial for HPV vaccine acceptability and uptake. 

However, the HPV vaccine is not the only vaccine marred with misinformation about Adverse Events Following Immunization. Other vaccines, such as Diphtheria, Tetanus, and Pertussis, and Measles, have suffered a similar fate [85]. Gains made are quickly lost amid misinformation, for example, in Ireland, groups of parents started sending out wrong information; fortunately, these were countered promptly [86]. A similar trend was noted in Senegal among healthcare workers [66]. Therefore, general messaging around vaccines improves understanding and increases end-user confidence across multiple stakeholders. 

Even though the cluster with strategies to “utilize financial strategies” was not frequently used and deemed to be of both low feasibility and importance by the initial expert consensus, access new funding and fund and contract for clinical innovation were deemed important though less feasible. Therefore, the non-feasibility of raising new funding in SSA may explain the presence of funders, such as Gavi, the vaccine alliance, Merck, Bill and Melinda Gates Foundation, GlaxoSmithKline, PATH, and others, that have made it possible to increase vaccination in many SSA countries by supporting the supply chain; therefore, partially, or completely removing the cost to the end users. However, concern remains whether the Gavi-supported countries will sustain the HPV vaccination program when they transition to self-financing due to the high cost of the vaccine and its implementation [87].

## 5. Limitations

Scoping reviews are ideal to determine the scope of literature on a given topic as we have established here; however, there are some inherent limitations. We did not synthesize the findings or assess the methodological quality and risk of bias for included studies, according to guidance by the JBI methodology for scoping reviews [38]; therefore, the findings should be used with caution when guiding policy statements. Additionally, the expert panel involved in developing the ERIC taxonomy was drawn from the US and North America; however, the original compilation was drawn from taxonomies developed in other contexts; therefore, the strategies are applicable in SSA [36]. Finally, we only included studies reported in English, and thus may have missed other important research reported in other languages.

## 6. Conclusions

This scoping review has mapped and provided a broad overview of implementation strategies used to increase HPV vaccination uptake in SSA using the ERIC compilation and reported research gaps unique to SSA. Implementation and sustainability of the HPV vaccination remain challenging, as shown by low global levels of HPV vaccine uptake, which is lowest in SSA. We present clearly defined discrete implementation strategies to ease replication and understanding of findings. Most strategies used were of high importance and feasibility, which indicates that they could be adapted within the SSA to increase HPV vaccine uptake for adolescent girls.

We found low availability or reporting of implementation strategies for national HPV vaccination rollout programs, even though 17 countries in SSA are currently implementing this program [8]. Most studies reported on demonstration projects and small studies, which served as a learning point before full scale up which was under the cluster of “use evaluative and iterative strategies”. Improved reporting of implementation strategies by national programs is needed to learn lessons from similar contexts considering that most SSA countries are now planning or are introducing national HPV vaccination programs. Furthermore, these studies should be well designed with clearly defined strategies for better evaluations and comparisons across studies and settings.

## Figures and Tables

**Figure 1 vaccines-11-01246-f001:**
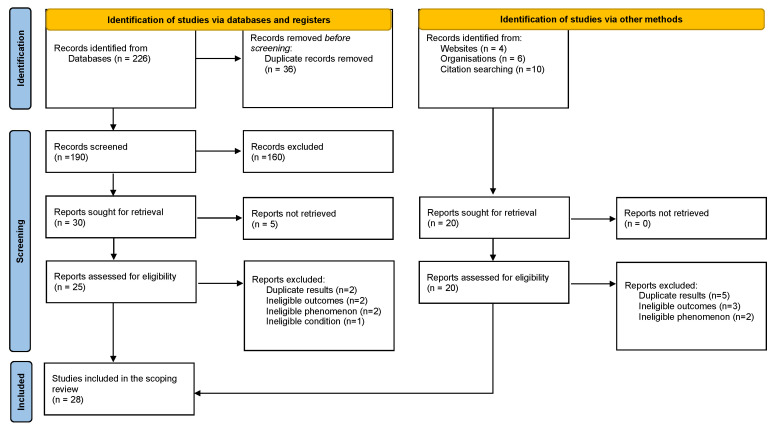
PRISMA flow chart adapted from http://www.prisma-statement.org/ accessed on 16 June 2023.

**Table 1 vaccines-11-01246-t001:** Characteristics of included articles.

First Author Name; Year [Reference]	Title of Article	Country of Study	Type of Data Source	Type of Program	Targeted Stakeholder	School Grade/Age of Girls	Funding Agency
Binagwaho, Agnes; 2012 [65]	Achieving high coverage in Rwanda’s national human papillomavirus vaccination program	Rwanda	Perspective	National	Policymakers, Non-governmental organizations, Donor community, Clergy, Teachers, Healthcare workers, parents, adolescent girls, community health workers	Grade 6 or 12 years	Merck
Casey, Rebecca M.; 2021 [66]	National introduction of HPV vaccination in Senegal: Successes, challenges, and lessons learnt	Senegal	Research—cross sectional	National	Policymakers, healthcare workers, nongovernmental organizations, parents, adolescent girls	9 years	The Gavi
Chigbu, Chibuike O.; 2017 [64]	The impact of community health educators on uptake of cervical and breast cancer prevention services in Nigeria	Nigeria	Research—pre- and post-design	Sub-national	Nurses	9–13 years	Out of pocket
Delany-Muretwe; 2018 [31]	Human papillomavirus vaccine introduction in South Africa: implementation lessons from an evaluation of the national school-based vaccination campaign	South Africa	Research—cross sectional	National program	District leaders, EPI Leaders, Healthcare workers, teachers, defense forces, nursing schools, politicians, journalists	Grade 4 or 9–13 years	The Gavi
Drokow, Emmanuel K.; 2021 [62]	The Impact of Video-Based Educational Interventions on Cervical Cancer, Pap Smear, and HPV Vaccines	Ghana	Research—pre- and post-design	Sub-national research	Parents	N/A	The Gavi
Engel, Danielle; 2021 [67]	Promoting adolescent health through integrated human papillomavirus vaccination programs: The experience of Togo	Togo	Evaluation	Demonstration in two districts	Technical partners, healthcare workers, adolescents, family members, community health workers	Grade 5 or 10 years	The Gavi
Galagan, Sean R.; 2013 [44]	Influences on parental acceptability of HPV vaccination in demonstration projects in Uganda and Vietnam	Uganda/Vietnam	Research—cross sectional	Demonstration program in two districts	Parents	Grade 5 or 10 years	The Gavi
Gallagher, Katherine E.; 2017 [45]	Lessons learnt from delivering HPV vaccine in 45 LMICs	45 LMICs (SSA countries included)	Research—Ecological study	Demonstration program	Policymakers, healthcare workers, non-governmental organizations, parents, adolescent girls	9–18 years	Variable
Jones, Amy; 2021 [53]	Using branded behavior change communication to create demand for the HPV vaccine among girls in Malawi: An evaluation of Girl Effect’s Zathu mini magazine	Malawi	Research—pre- and post-design	Sub-national research	Girls, parents, and influencers	9 years	N/A
Kabakama, Severin; 2016 [47]	Social mobilization, consent, and acceptability: A review of human papillomavirus vaccination procedures in low- and middle-income countries	37 LMICs (SSA countries included)	Literature review	LMICs—National and sub-national	Girls, parents, and influencers	Variable, routine	Variable
Ladner, Joe; 2016 [50]	Experiences and lessons learned from 29 HPV vaccination programs implemented in 19 low- and middle-income countries, 2009–2014	LMICS (SSA countries included)	Perspective	Demonstration program	parents, community, government, multiple	9–13 years	Global Access Program
Ladner, Joel; 2012 [46]	Assessment of implementation of HPV vaccination programs in eight of the lowest-income countries (SSA—Cameroon and Lesotho)	LMICs—Cameroon and Lesotho included	Program evaluation	Demonstration program	Policymakers, educators, healthcare workers	9–13 years	Global Access Program
LaMontagne, Scott D.; 2021 [69]	HPV vaccination coverage in three districts in ZIMBABWE following the national introduction of a schedule of 0 to 12 months among girls aged 10 to 14 years	Zimbabwe	Research—cross sectional	National program	Parents	Multi-age cohort (10–14 years)	The Gavi
LaMontagne, D. Scott; 2011 [43]	Human papillomavirus vaccine delivery strategies that achieved high coverage in low- and middle-income countries	LMICs/Uganda	Research—cross sectional	Demonstration in two districts	Healthcare workers, community mobilizers, Parents, Adolescent girls	9–14 years	Merck and Co./Gloxosmithline
Moodley, Indres; 2013 [51]	High uptake of Gardasil vaccine among schoolgirls aged 9–12 years participating in an HPV vaccination demonstration project in KwaZulu-Natal, South Africa	South Africa	Research—cross sectional	Demonstration, one province	Nurses, teachers, religious, district health, district education health	9–12 years	Research grant
Mpuru, Alex; 2021 [56]	National introduction of human papillomavirus (HPV) vaccine in Tanzania: Programmatic decision-making and implementation	Tanzania	Report	National	EPI TWG, policymakers, politicians, religious leaders, healthcare workers, community health volunteers, school personnel, media personnel	14 years	The Gavi
Msyamboza, Kelias P.; 2017 [54]	Implementation of human papillomavirus vaccination demonstration project in Malawi: Successes and challenges	Malawi	Research—cross sectional	Demonstration program in two districts	Healthcare workers, policymakers, parents	Grade 4 or 9–13 years out-of-school	The Gavi
Mugisha, Emmanuel; 2015 [57]	Feasibility of delivering HPV vaccine to girls aged 10–15 years in Uganda	Uganda	Research—cross sectional	Demonstration project in two districts	District leaders, EPI Leaders, Healthcare workers	Grade 5 or 10 years	Glaxosmithkline Biologicals SA
Program for Approriate Technology in Health; 2011 [58]	HPV Vaccination in Africa, lessons learnt from a pilot program in UGANDA	Uganda	Report	Demonstration project in two districts	Policymakers, planners, teachers, community members, adolescent girls, parents	Grade 5 or 10 years	Glaxosmithkline biologicals SA
Raesima, Mmakgomo M.; 2015 [59]	Human Papillomavirus Vaccination Coverage Among School Girls in a Demonstration Project—Botswana, 2013	Botswana	Report	Demonstration program, one district	Teachers, parents, adolescent girls, and healthcare workers	Grades 4 to 6 or >9 years	Pink Ribbon Red Ribbon
Snyman, Leon; 2015 [52]	The Vaccine and Cervical Cancer Screen project 2 (VACCS 2): Linking cervical cancer screening to a two-dose HPV vaccination schedule in the South West District of Tshwane, Gauteng, South Africa	South Africa	Research—cross sectional	Sub-national Gauteng and Western provinces	Adolescents and female parents	Grades 4 to 7 or 9 years	Manufacturer donation
Soi, Catherine; 2018 [63]	Human papillomavirus vaccine delivery in Mozambique: identification of implementation performance drivers using the Consolidated Framework for Implementation Research (CFIR)	Mozambique	Evaluation	Demonstration—three regions	Healthcare workers, policymakers, community leaders	Grade 4	The Gavi
Tsu, Vivien; 2021 [48]	National implementation of HPV vaccination programs in low-resource countries: Lessons, challenges, and future prospects	LMICs (SSA countries included)	Literature review	National programs	Journalists, healthcare workers, teachers, parents, community leaders, adolescent girls	Variable, routine	Variable
Wamai, Richard; 2012 [60]	Assessing the effectiveness of a community-based sensitization strategy in creating awareness about HPV, cervical cancer, and HPV vaccine among parents in North West Cameroon	Cameroon	Research—cross sectional	Regional—one district	Parents	9–13 years	Church donation
Watson, Deborah-Jones; 2012 [55]	Human papillomavirus vaccination in Tanzanian schoolgirls: Cluster-randomized trial comparing two vaccine-delivery strategies	Tanzania	Research—cluster randomized trial	Regional research in two districts	Teachers, Adolescents, Parents	Grade 6 or 12 years	Wellcome trust
Whitworth, Hilary; 2021 [49]	Adolescent Health Series: HPV infection and vaccination in sub-Saharan Africa: 10 years of research in Tanzanian female adolescents—narrative review	SSA	Literature review	Regional—SSA	Policymakers, teachers, healthcare workers, parents, adolescent girls	N/A	N/A
World Health Organisation; 2013 [68]	Human papillomavirus (HPV) vaccine introduced in Zambia	Zambia	Report	Demonstration in three districts	Policymakers, teachers, healthcare workers, parents, adolescent girls	Grade 4 or 10 years	The Gavi
World Health Organisation; 2015 [61]	First Lady launches the HPV vaccine project for the prevention of cervical cancer	The Gambia	Report	Demonstration project in two districts	Political figures, policymakers, parents, adolescent girls	9–13 years	The Gavi

**Table 2 vaccines-11-01246-t002:** Implementation strategies used by cluster.

Implementation Strategies Cluster	Total No. of Strategies within Each Cluster	Strategies Used per Cluster across 28 Papers No. (%)	Total No. of Times Strategy Identified by Scoping Review
1. Use evaluative and iterative strategies	10	100% (10/10)	113
2. Provide interactive assistance	4	100% (4/4)	51
3. Adapt and tailor to context	4	75% (3/4)	45
4. Develop stakeholder interrelationships	17	94% (16/17)	164
5. Train and educate stakeholders	11	100% (11/11)	131
6. Support clinicians	5	80% (4/5)	26
7. Engage consumers	5	100% (5/5)	65
8. Utilize financial strategies	9	44% (4/9)	27
9. Change Infrastructure	8	63% (5/8)	46

## Data Availability

Data are available in the manuscript text and supplementary materials.

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
