# Peer review of "Implementation Strategies Used to Increase Human Papillomavirus Vaccination Uptake by Adolescent Girls in Sub-Saharan Africa: A Scoping Review"

_vaccines, 2023, doi:10.3390/vaccines11071246_

Round 1

Reviewer 1 Report

The authors have tried their best for this review papers. I have the following observation to improve the readability of the paper:

1. The similarity content is very high from the following sources ( https://journals.plos.org/plosone/article?id=10.1371%2Fjournal.pone.0267617

2. Figure in page 5 doesn't have caption and citation/refrerences

3. In line number 257 , the author say "Table 3 shows the distribution of the identified implementation strategies within 257 the 9 ERIC clusters, and frequency of their use within the articles", but I couldn't find the table 3 in the text.

4. In Table 2 : Implementation strategies cluster* , I didn't understand hat is "*" shows there.

5.Both the discussion and the conclusion parts of the paper have been written quite nicely.

Author Response

REVIEWER 1

  1. The similarity content is very high from the following sources ( https://journals.plos.org/plosone/article?id=10.1371%2Fjournal.pone.0267617

Thank you, this is because the referenced article is the protocol written prior to conducting this scoping review; however, we have paraphrased some sentences within the introduction and methods sections of this scoping review paper to reduce similarity.

  1. Figure in page 5 doesn't have caption and citation/reference

Thank you, the caption has been added as: Figure 1: PRISMA flow chart, line 228

  1. In line number 257, the author say "Table 3 shows the distribution of the identified implementation strategies within 257 the 9 ERIC clusters, and frequency of their use within the articles", but I couldn't find the table 3 in the text.

Thank you, this should read as 'Table 2’ and not ‘Table 3,’ this has been corrected.

  1. In Table 2: Implementation strategies cluster*, I didn't understand hat is "*" shows there.

Thank you, that was in error and it has since been deleted

5.Both the discussion and the conclusion parts of the paper have been written quite nicely.

Thank you

Author Response

REVIEWER 2:

I don't find this article to a good fit for Vaccines. Although it might to be some relevance but most conclusion made/listed are very intuitive.

Thank you for the comment. The manuscript has been improved to make it a good fit for vaccines.

Reviewer 3 Report

The present scoping review reports about implementation strategies used to increase HPV vaccination uptake by adolescent girls in SSA. The review used ERIC taxonomy to identify implementation strategies in the context of SSA countries to overcome the barriers associated with HPV vaccine uptake in adolescent girls.

Following are the comments to highlight certain limitations after review of the paper;

1.     Line no 63 – Kindly define older population, preferably mention the age group.

2.     Line no 82-87 – WHO 90-70-90 target for WHO cervical cancer elimination strategy by 2030 is mentioned in lines. However, the sentence formation and explanation gives a wrong idea about facts. Authors mentioned that 90% of girls aged 15 years…, the sentence gives an idea that 90% of girls who attained age 15 must be fully vaccinated, while the WHO target implies that 90% of girls by the age of 15 should be fully vaccinated. Same goes with second target of 70% - authors stated that women aged 35-45 years screened at least twice while the WHO target states that 70% of women by age 35 should be screened with a high performance test and again screened at 45 years of age. In the third target authors mentioned that 90% of pre cancers and cancerous lesions treated appropriately while WHO targets states that 90% of women with pre cancer treated and 90% with invasive carcinoma managed properly. Authors are requested to clarify the same.

3.     Flowchart, Section 3.1 – What are the exclusion criteria for records after second step? The same has not been mentioned. Kindly add and clarify.

4.     Flowchart, Section 3.1 – Out of 28 records selected, how many of them were entries from grey literature?

5.     What is the idea behind incorporating grey literature in the review? Grey literature is not primarily peer reviewed. Did it pose any implications on the records selected thereby affecting quality of data in the review?

6.     Line no. 218-220 – Most of the included articles…. The complete data sets are not mentioned. Kindly complete the sentence and mention all the data sets.

7.     Line no. 221-224 – Kindly check the data mentioned in these lines. Authors have selected 28 records while datasets mentioned in these lines are 31 records.

8.     Table 1 – Kindly mention the funding in table to give concise idea about financial barriers for vaccine uptake and access new funding strategies.

9.     Authors are suggested to shed a light on vaccine cost as a barrier in uptake of vaccines and the role it plays in implementation strategies.

10.  Line no. 259 – Kindly define implementation strategy cluster 1 – Use and evaluative iterative strategies. Kindly explain what strategies it encompasses.

11.  Line no 252 & 271 – Kindly check the number of records in access new funding strategy. The data is different in both the lines.

12.  All the implementation strategies used by cluster are not discussed extensively in the discussion section with possible effects on SSA countries. Authors are suggested to discuss the same and the possible outcomes of addressing the barriers associated with them.

13.  Kindly discuss about barriers to vaccine uptake with respect to implementation strategies identified.

General Comment - Why have authors not enhanced the utility of the compilation by linking it to a conceptual framework like CFIR (Consolidated Framework for Implementation Research). One of the possible limitations of ERIC is the absence of conceptual model to generate data. Authors could have used ERIC-CFIR matching tool to generate a robust data and idea about implementation strategies and barriers involved. Kindly comment on the same.

Acceptable, minor editing of English language required.

Author Response

REVIEWER 3

  1. Line no 63 – Kindly define older population, preferably mention the age group.

Thank you, those 21 years and older. This has been corrected. See Lines 63-64.

  1. Line no 82-87 – WHO 90-70-90 target for WHO cervical cancer elimination strategy by 2030 is mentioned in lines. However, the sentence formation and explanation gives a wrong idea about facts. Authors mentioned that 90% of girls aged 15 years…, the sentence gives an idea that 90% of girls who attained age 15 must be fully vaccinated, while the WHO target implies that 90% of girls by the age of 15 should be fully vaccinated. Same goes with second target of 70% - authors stated that women aged 35-45 years screened at least twice, while the WHO target states that 70% of women by age 35 should be screened with a high-performance test and again screened at 45 years of age. In the third target authors mentioned that 90% of pre cancers and cancerous lesions are treated appropriately, while WHO targets states that 90% of women with pre cancer treated and 90% with invasive carcinoma managed properly. Authors are requested to clarify the same.

Thank you, this has been corrected. See line 90-94

  1. Flowchart, Section 3.1 – What are the exclusion criteria for records after second step? The same has not been mentioned. Kindly add and clarify.

Thank you, these records were screened by title or abstract as shown in the preceding box. Studies that did not meet the inclusion criteria based on title or abstract were excluded.

  1. Flowchart, Section 3.1 – Out of 28 records selected, how many of them were entries from grey literature?

Three  entries

  1. What is the idea behind incorporating grey literature in the review? Grey literature is not primarily peer reviewed. Did it pose any implications on the records selected thereby affecting quality of data in the review?

Thank you, according to JBI methodology for scoping reviews which guided this review, “A scoping review can include any and all types of literature (e.g., primary research studies, systematic reviews, meta-analyses, letters, guidelines, websites, blogs).” https://journals.lww.com/jbisrir/Fulltext/2020/10000/Updated_methodological_guidance_for_the_conduct_of.4.aspx

Therefore, based on our research question and objectives, it was appropriate that we include grey literature.Further, grey literature is increasingly being used in different types of reviews to minimize publication bias. https://onlinelibrary.wiley.com/doi/full/10.1111/ijmr.12102

The addition of grey literature to our included studies has not affected the quality of our review.

  1. Line no. 218-220 – Most of the included articles…. The complete data sets are not mentioned. Kindly complete the sentence and mention all the data sets.

Thank you, this has been corrected. Perspectives have been added. Line 237.

  1. Line no. 221-224 – Kindly check the data mentioned in these lines. Authors have selected 28 records while datasets mentioned in these lines are 31 records.

Thank you, this has been corrected, there was an error. Lines 241-242

  1. Table 1 – Kindly mention the funding in table to give concise idea about financial barriers for vaccine uptake and access new funding strategies.

Table 1 has been updated

  1. Authors are suggested to shed a light on vaccine cost as a barrier in uptake of vaccines and the role it plays in implementation strategies.

This has been addressed in lines 393-396

  1. Line no. 259 – Kindly define implementation strategy cluster 1 – Use and evaluative iterative strategies. Kindly explain what strategies it encompasses.

Thank you, cluster 1 encompasses evaluative strategies as detailed in Supplementary Table 4 and listed below. Supplementary Table 4 lists the clusters and the strategies encompassed in each.

Assess for readiness and identify barriers and facilitators, Audit and provide feedback, Purposefully re-examine the implementation, Develop and implement tools for quality monitoring, Develop and organise quality monitoring systems, Develop a formal implementation blueprint, Conduct local needs assessment, Stage implementation scale-up, Obtain and use patients/consumers and family feedback, Conduct cyclical small tests of change 

  1. Line no 252 & 271 – Kindly check the number of records in access new funding strategy. The data is different in both the lines.

The correct data is 21, thank you.

  1. All the implementation strategies used by cluster are not discussed extensively in the discussion section with possible effects on SSA countries. Authors are suggested to discuss the same and the possible outcomes of addressing the barriers associated with them.

Thank you, edits have been made to the abstract line 36 and discussion sections lines 307-325, 326-338, 348-362, 364-376, 393-400, 430-433. However, discussing addressing barriers associated with these strategies is beyond the scope of this scoping review as it was not the objective of this scoping review to extract specific barriers associated with the identified strategies

  1. Kindly discuss about barriers to vaccine uptake with respect to implementation strategies identified.

Thank you. However, this is outside the scope of this scoping review. We intend to conduct a systematic review where we will delve into identifying barriers, facilitators and effective strategies to HPV vaccination in SSA, and we will definitely consider this suggestion

General Comment - Why have authors not enhanced the utility of the compilation by linking it to a conceptual framework like CFIR (Consolidated Framework for Implementation Research). One of the possible limitations of ERIC is the absence of conceptual model to generate data. Authors could have used ERIC-CFIR matching tool to generate a robust data and idea about implementation strategies and barriers involved. Kindly comment on the same.

Thank you for this recommendation, and it is most welcome.

We did not consider the CFIR-ERIC matching tool as the aim of this scoping review was not to identify barriers or facilitators to implementing the HPV vaccination program. Our primary goal was to understand and map the available evidence on implementation strategies as a starting point to explore available data - see lines 115-118. Additionally, the authors of the CFIR-ERIC matching tool recommended more research on the tool due to considerable heterogeneity among the respondents suggesting relatively few consistent relationships between the CFIR Barriers and ERIC implementation strategies: https://cfirguide.org/choosing-strategies/

Reviewer 4 Report

This scoping review is very well developed and structured. It is based on a solid, well-explained methodology.

The results are complete and well detailed.

I have only one comment: include the inclusion criteria section in the method section of the study.

Congratulations to the authors for their work.

Author Response

REVIEWER 4

This scoping review is very well developed and structured. It is based on a solid, well-explained methodology. The results are complete and well-detailed.

Thank you

I have only one comment: include the inclusion criteria section in the method section of the study.

Thank you, the inclusion criteria have been moved to the methods section

Congratulations to the authors for their work.

Thank you

Reviewer 5 Report

Thank you for the opportunity to review this interesting manuscript. The human papillomavirus (HPV) is sexually transmitted and infects approximately 75% of sexually active people early in their sexual life. Despite the HPV vaccine being available for about 15 years, dose completion remains as low as 20% in sub-Saharan African countries implementing the vaccination program compared to 77% in Australia and New Zealand. It is a good job but I have some suggestion to improve this manuscript. 

Firstly, I reccomend to modify the flow chart and use The PRISMA DIAGRAM, available here http://prisma-statement.org/prismastatement/flowdiagram.aspx?AspxAutoDetectCookieSupport=1. 

I suggest to order the papers in the table by year or alphabetic order.

In the discussion I think that can be useful to disccuss strategies to offer HPV vaccination in other healtcare context. Here some references "Sinopoli A, Baccolini V, Di Rosa E. Killing Two Birds with One Stone: Is the COVID-19 Vaccination Campaign an Opportunity to Improve Adherence to Cancer Screening Programmes? The Challenge of a Pilot Project in a Large Local Health Authority in Rome. Vaccines. 2023; 11(3):523. https://doi.org/10.3390/vaccines11030523"

Minor editing of English language required.

Author Response

REVIEWER 5

Firstly, I recommend to modify the flow chart and use The PRISMA DIAGRAM, available here http://prisma-statement.org/prismastatement/flowdiagram.aspx?AspxAutoDetectCookieSupport=1. 

This has been done, thank you.

I suggest to order the papers in the table by year or alphabetic order.

This has been done, thank you. Papers have been ordered alphabetically in table 1.

In the discussion I think that can be useful to discuss strategies to offer HPV vaccination in other healthcare context. Here some references "Sinopoli A, Baccolini V, Di Rosa E. Killing Two Birds with One Stone: Is the COVID-19 Vaccination Campaign an Opportunity to Improve Adherence to Cancer Screening Programmes? The Challenge of a Pilot Project in a Large Local Health Authority in Rome. Vaccines. 2023; 11(3):523. https://doi.org/10.3390/vaccines11030523"

Thank you for this recommendation, strategies to offer the HPV vaccine has been discussed under the cluster change infrastructure and specifically under the strategy 'change service sites' from health facilities to schools and integrated with other services. Lines 355-364